# Use of Antibiotics among Residents Living Close to Poultry or Goat Farms: A Nationwide Analysis in The Netherlands

**DOI:** 10.3390/antibiotics10111346

**Published:** 2021-11-04

**Authors:** Inge Roof, Wim van der Hoek, Lisette Oude Boerrigter, Cornelia C. H. Wielders, Lidwien A. M. Smit

**Affiliations:** 1National Institute for Public Health and the Environment, P.O. Box 1, 3720 BA Bilthoven, The Netherlands; wim.van.der.hoek@rivm.nl (W.v.d.H.); lisette.oudeboerrigter@gmail.com (L.O.B.); lieke.wielders@rivm.nl (C.C.H.W.); 2Institute for Risk Assessment Sciences, Utrecht University, Yalelaan 2, 3584 CM Utrecht, The Netherlands; L.A.Smit@uu.nl

**Keywords:** antibiotic use, pneumonia, livestock, human, poultry, goat

## Abstract

Prior regional studies found a high risk of pneumonia for people living close to poultry and goat farms. This epidemiological study in the Netherlands used nationwide antibiotic prescription data as a proxy for pneumonia incidence to investigate whether residents of areas with poultry and goat farms use relatively more antibiotics compared to areas without such farms. We used prescription data on antibiotics most commonly prescribed to treat pneumonia in adults and livestock farming data, both with nationwide coverage. Antibiotic use was expressed as defined daily doses per (4-digit Postal Code (PC4) area)-(age group)-(gender)-(month) combination for the year 2015. We assessed the associations between antibiotic use and farm exposure using negative binomial regression. The amoxicillin, doxycycline, and co-amoxiclav use was significantly higher (5–10% difference in use) in PC4 areas with poultry farms present compared to areas without, even after adjusting for age, gender, smoking, socio-economic status, and goat farm presence. The adjusted models showed no associations between antibiotic use and goat farm presence. The variables included in this study could only partly explain the observed regional differences in antibiotic use. This was an ecological study that precludes inference about causal relations. Further research using individual-level data is recommended.

## 1. Introduction

In the Netherlands, livestock farming has intensified in the past decades [1]. Currently, with a population of 17 million people, the Netherlands accommodates more than 100 million broilers and laying hens, 12.2 million pigs, 3.8 million cows and veal calves, 0.9 million sheep, and 0.6 million goats [2]. There is an increasing concern about potential health risks related to this high livestock density in close proximity to residential areas. Previous studies have shown the significant contribution of agriculture to air pollution, with large quantities of particulate matter (PM), ammonia, and endotoxins emitted by livestock farms [3,4,5,6]. Farm-related exposures can lead to airway obstruction and increased respiratory symptoms, especially in chronic obstructive pulmonary disease patients [7,8]. Additionally, air pollution exposure may increase susceptibility to respiratory infections by chronic airway inflammation and reduced host defense function [9,10].

Large epidemiological studies, mainly in the United States and western Europe, have shown associations between living in livestock-dense areas and adverse respiratory health effects, including increased respiratory symptoms and decreased lung function [7,8,11,12,13]. In addition, an increased risk of community-acquired pneumonia (CAP) among adults living in close vicinity (1–2 km range) of poultry and goat farms has been observed [14,15,16,17,18,19,20]. The PM emitted from poultry farms has been proposed as a plausible explanation for the association between poultry farms and CAP. However, not all prior studies in the Netherlands pertaining to poultry farms and CAP found consistent, significant results [14,21]. For goat farms, the association with CAP has been more consistent over a long period, from 2009 to 2016 [17,22,23]. However, goat farms emit much lower PM levels than poultry farms, and possible underlying biological mechanisms remain unknown. The research on livestock exposure and respiratory health effects also shows limitations, because research has only been performed at a regional level. Furthermore, the primary data sources were respiratory health questionnaires and electronic medical records from general practitioners, who generally diagnose CAP based on clinical criteria without radiological or microbiological analysis [24].

Despite inconsistencies and lack of information on causal agents, the studies supporting an association between livestock farms and CAP have had an immediate policy impact in the Netherlands. Nine of the twelve provinces have stopped issuing building permits for new and existing goat farms, and poultry farms are obliged to reduce PM emissions. Considering this immediate impact, additional research at a national level is essential to better understand previous findings.

The majority of pneumonia patients in the Netherlands are treated with antibiotics in primary care [24,25]. If the CAP incidence is higher in poultry and goat farm-dense areas, we would also expect a higher use of antibiotics specifically prescribed for pneumonia in these areas. Studying prescription data of antibiotics used to treat pneumonia can therefore provide more insight in the association between livestock and CAP or other infectious diseases for which common antibiotics are prescribed. Moreover, research on spatial variation in antibiotic consumption within the Netherlands is scarce. Previous studies in the Netherlands and other countries have identified some factors that can influence antibiotic consumption, such as socio-economic status (SES), age, and gender distribution [26,27,28,29,30,31]. The exact causes of the differing regional prescription patterns, and to what extent livestock exposure plays a role, is unknown. In this study, outpatient antibiotic prescription and poultry and goat farming data were used, both with national coverage, to assess whether residents of areas with poultry and goat farms use relatively more antibiotics compared to residents of areas without poultry and goat farms.

## 2. Results

### 2.1. Population Characteristics

Data on antibiotic use, poultry and goat farm presence, and population size were collected for 4024 4-position Postal Code (PC4) areas in the Netherlands. After removal of PC4 areas that were highly urbanized (score 1 or 2), that contained fewer than 100 inhabitants, or that showed missing data on SES, 2586 PC4 areas were included. The number of inhabitants per PC4 area ranged from 210 to 21,690. Each Unique Data Point (UDP) represented a unique gender-age-month combination, meaning that per PC4 area, 384 UDPs (2 gender categories × 16 age categories × 12 months) were possible. In total, 993,024 UDPs were included in the final dataset.

Age and gender were similarly distributed between the UDPs with and without poultry or goat farms (Table 1). The SES score ranged from −6.65 to 3.06 with a mean score of 0.43. SES scores were lower for UDPs with poultry farms than for UDPs without poultry farms. Smoking percentages within a PC4 area ranged from 11% to 30%. Smoking prevalence was lower in UDPs with poultry or goat farms than in UDPs without farms. The average number of farms per PC4 area varied between 0 and 5.5 for poultry and between 0 and 1.5 for goats. There were more PC4 areas with poultry farms present (*n* = 1052) compared to goat farms (*n* = 486).

### 2.2. Data on Antibiotic Use

Human antibiotic prescription data for 2015 were collected for doxycycline, amoxicillin, co-amoxiclav, and nitrofurantoin. The total number of prescriptions was highest for amoxicillin (531,962) and lowest for co-amoxiclav (430,703). A trend of increasing antibiotic use with an increasing age was visible for all antibiotics (Table 2). Differences in mean Defined Daily Doses (DDD) between males and females were present, most predominantly for nitrofurantoin. The mean DDD per 1000 inhabitants decreased with increasing SES. For all antibiotic types, the mean DDD per 1000 inhabitants was higher in the PC4 areas with poultry or goat farms, but this was most pronounced for doxycycline (Table 2).

### 2.3. Maps

Visual inspection of the maps showed remarkable regional differences in antibiotic use with high use patterns in the south-west and north-east of the country (Figure 1). These are areas with a generally low SES (Appendix A). Poultry farms were mostly clustered in the south-east and in the middle of the Netherlands, whereas for goat farms less clustering was present (Appendix A). No clear spatial association was visible between antibiotic use and the presence of poultry or goat farms.

### 2.4. Main Analyses

Crude two-by-two associations were evaluated for the different antibiotics with dichotomous DDD (DDD = 0 or DDD > 0) as outcome variable and presence of poultry and goat farms as risk factors. Table 3 shows that, for all types of antibiotics, the use was higher among the UDPs with poultry or goat farms present compared to the UDPs without farms.

The ratios resulting from the negative binomial regression models are presented in Table 4. For amoxicillin, doxycycline, and co-amoxiclav, the crude models showed that the ratios of antibiotic use in the UDPs with poultry farms were 1.08 (95% CI: 1.05–1.11), 1.15 (95% CI: 1.11–1.20), and 1.08 (95% CI: 1.05–1.12), respectively, compared to the UDPs without poultry farms. These ratios indicated that the associated antibiotic use of amoxicillin, doxycycline, and co-amoxiclav was 8%, 15%, and 8% higher in the UDPs with poultry farms. In the adjusted models, with correction for differences in age, gender, SES, smoking, and presence of goat farms, the associations remained present; however, the adjusted ratios were closer to 1: amoxicillin 1.05 (95% CI: 1.02–1.08), doxycycline 1.10 (95% CI: 1.06–1.14), and co-amoxiclav 1.05 (95% CI: 1.02–1.08). In the crude model for nitrofurantoin, the ratio of antibiotic use in the UDPs with poultry farms was 1.06 (95% CI: 1.02–1.11). This effect diminished to a ratio of 1.00 (95% CI: 0.96–1.04) in the adjusted negative binomial regression model. In the crude models with goat farm presence as exposure variable, only a statistically significant ratio was found for doxycycline (1.06, 95% CI: 1.01–1.12) and nitrofurantoin (1.09, 95% CI: 1.03–1.15). In the adjusted model, with correction for demographics, smoking, and poultry farm presence, no clear association between goat farm presence and the four types of antibiotics was found (Table 4).

### 2.5. Sensitivity Analyses

Stratification of the negative binomial regression models by season gave similar results as in the main analysis. Neither the crude nor adjusted ratios differed between the winter months and other months (Appendix A). Restricting the analysis to the livestock dense area in the Netherlands did not alter the effect estimates, except the estimates of doxycycline use and poultry farm presence were lower than in the main analyses: crude ratio 1.07 (95% CI: 1.02–1.13) and adjusted ratio 1.05 (95% CI: 1.00–1.10) (Appendix A).

## 3. Discussion

The present study shows spatial associations between the presence of poultry farms and the use of three common antibiotics. These associations remained after adjusting for age, gender, SES, smoking, and the presence of goat farms, suggesting an independent effect of poultry farms. The association was strongest for doxycycline, with an adjusted ratio of 1.10, indicating that the doxycycline use was 10% higher in the UDPs with poultry farms than in UDPs without poultry farms.

The higher use of antibiotics prescribed for CAP in inhabitants of areas with poultry farms corresponds with the increased risk of CAP around poultry farms that was shown in the studies in the Netherlands and the US [15,17,19]. Dutch studies show a consistent high risk of pneumonia in individuals living around goat farms [32], but in the adjusted models of the present study, no association was found between the presence of goat farms and antibiotic use. This may have to do with imprecision of our exposure classification on a PC4 level, which comprises large residential areas. The previous associations between goat farms and pneumonia, based on exact residential locations, were particularly present for residents at shorter distances (500–1000 m). By using PC4 level, our exposure estimation is less precise and could be misclassified. Exposure misclassification for poultry farms might be smaller, as poultry farms are more prevalent than goat farms in the Netherlands and often multiple poultry farms per PC4 area are present.

The antibiotic use in this study was expressed in DDDs. This technical unit is customary in international literature on antibiotic use, making the results of this study comparable to other literature. In this study, the antibiotics most commonly used to treat pneumonia were selected and the number of DDDs correctly indicates the number of outpatient antimicrobial prescriptions at national level. We used the antibiotic prescription data as another proxy for pneumonia incidence; however, the selected antibiotics are also used to treat other infections. In fact, amoxicillin, the first line antibiotic for pneumonia, is among the most commonly prescribed antibiotics in the Netherlands [33]. This indicates that the outcome variable selected in this study does not only represent the occurrence of CAP but also other respiratory and non-respiratory bacterial infections, which might also explain the weak associations found. For example, Chronic Obstructive Pulmonary Disease might act as a confounder, because previous research showed that living close to livestock farms is associated with a lower probability of medication dispensing for obstructive airways diseases [34], although these medication types mostly included inhalants and not antibiotics.

The influence of SES on antibiotic use has already been shown in multiple studies [26,28,30]. Visualization of antibiotic use in our maps showed remarkable regional differences, that generally coincided with difference in SES. We adjusted for SES of each PC4 area in our negative binomial regression models by using the social status score. This score is just one variable combining educational level, income, and position on the labor market of the inhabitants of each PC4 area. The status score variable might not be a complete representation of the actual SES. Residual confounding by SES can therefore not be excluded. Moreover, other unmeasured factors, such as regional differences in health-seeking behavior or prescribing practices by medical practitioners, might also contribute to residual confounding of the results.

Our hypothesis was that inhabitants of areas with livestock farms use relatively more antibiotics commonly prescribed for pneumonia, as a result of a higher CAP incidence in these areas. Nitrofurantoin was intentionally included as a control. Since this antibiotic is exclusively prescribed for urinary tract infections [35], no relation between livestock farms and nitrofurantoin was expected. In this study, the adjusted ratios of nitrofurantoin use and poultry and goat farms presence appeared to be weaker or had a confidence interval containing 1.00, compared to the other three antibiotic types. The incidence of urinary tract infection is much higher among females than males [36] and this explains the large gender difference in mean DDD per 1000 inhabitants (male: 19.5, female: 105.8, Table 2). However, restriction of the negative binomial regression analysis to females did not alter the ratios and interpretations (Appendix A).

CAP incidence generally shows seasonal variation with peaks during the winter months. In this study, we found similar crude and adjusted ratios for antibiotic use when we stratified the analysis between winter months and other months. In spring and summer, atypical causative agents of CAP including zoonotic agents are responsible for a large proportion of CAP episodes [37]. If a specific pathogen transmitted from the farms to humans causes the elevated pneumonia risk and the pathogen itself has a specific seasonality, we would expect to see stronger effect estimates in spring and summer. If the causal mechanism involves a predisposing factor, e.g., PM or endotoxin emissions, the effect of living close to goat and poultry farms might remain during both seasons. A more detailed exploration of seasonal effects is recommended in future studies regarding pneumonia and goat or poultry farm exposure.

A strong point of the study is that we had nationwide data sources on antibiotic prescriptions, livestock farming, population characteristics, SES, and smoking behavior, which enabled us to study the relation between livestock farming and antibiotic use and adjust for confounders at the national level. First exploration of the negative binomial regression models showed clustering of characteristics between PC4 areas, but by including PC4 as a random effect in all models, the analyses were corrected for this potential clustering.

However, the major limitation of our cross-sectional and ecological study design is that no causality can be derived between the exposures and outcome. Because aggregated data were used, the results cannot be translated to the individual level. Additionally, we classified exposure on a PC4 level, which is less accurate than on full six position postal code level that was used in the previous studies regarding CAP and livestock exposure. The measurement error in the level of exposure possibly introduced non-differential exposure misclassification and thereby biased the associations towards the null. Due to privacy restrictions, it was not possible to collect the data on individual or on full postal code level. Even though no individual data were used, the dataset still contained a very large number of observations. When working with such a large amount of data, the chance of getting statistically significant results for small, irrelevant effect sizes is relatively high. Hence, we chose to present 95% CIs of the ratios from the negative binomial regression models rather than p-values.

## 4. Materials and Methods

### 4.1. Study Design

We conducted a cross-sectional study using aggregated data at national level in the Netherlands for 2015. Data on antibiotic use, livestock farming, and additional factors such as population size, SES, urbanization level, and smoking were linked at 4-position postal code (PC4) level. Since anonymous, aggregated data were used, permission from a Medical Ethical Committee was not necessary according to Dutch legislation.

### 4.2. Data on Livestock Farming

Data on exact location of poultry and goat farms in 2015 in the Netherlands were provided by the Netherlands Enterprise Agency. These data only include the location, type of farm, and the number of animals and do not contain information about specific characteristics of farms. The presence of poultry or goat farms, with a minimum number of 100 and 50 animals, respectively, within a range of 1 km of residential addresses were used as primary exposure variables. A range of 1 km was selected based on the results of prior studies [17,18]. In the Netherlands, postal codes consist of 4 digits (PC4) followed by two upper case letters (6 position postal code or PC6). The PC4 represents a region up to neighborhood level, where PC6 reflects a street or specific part of a street. For this study, we calculated the presence of the farms by first placing a circle with a 1 km radius over all full PC6 centroids and determining the number of poultry and goat farms within each PC6 area. Subsequently, the population weighted average of all PC6 areas was calculated to determine the average number of farms per PC4 area. Based on that average number of farms, the presence of poultry and goat farms within a PC4 area was coded as a separate dichotomous variable (>0 farms on average coded as yes, 0 farms coded as no).

### 4.3. Antibiotic Prescription Data

We obtained antibiotic prescription data from the Drug Information System (GIP) of the National Health Care Institute (in Dutch: GIP/College voor zorgverzekeringen, Zorginstituut Nederland). The GIP database systematically collects data from health insurers that together cover 96% of the extramural drug use of the Dutch population. GIP has extrapolated the data to achieve coverage of the entire Dutch population. The database included data on drugs that are prescribed by general practitioners and medical specialists, and that are dispensed by pharmacists, dispensing general practitioners, and other outlets. Drug prescriptions for patients admitted to hospitals or nursing homes are not included. Antibiotic prescriptions are registered according to the Anatomical Therapeutic Chemical classification system and expressed in Defined Daily Doses (DDD) [38], number of users and number of prescriptions. For privacy reasons, data on antibiotic use are only available by (PC4)-(five year age group)-(gender)-(month) combination. This resulted in a dataset with data points that all contained a unique (PC4)-(age group)-(gender)-(month) combination and that are called Unique Data Points (UDPs).

We restricted our analyses to the antibiotics amoxicillin, doxycycline, and co-amoxiclav (amoxicillin with clavulanic acid) since these drugs are most commonly, yet not specifically, prescribed for pneumonia in adults, based on prevailing professional guidelines of the Dutch College of General Practitioners (NHG) and the Dutch working Group on Antibiotic Policy (SWAB) [24,39]. Nitrofurantoin was included as a control antibiotic, since this drug is exclusively used for urinary tract infections. Only data from persons aged 15 years and older were included as other antibiotics are recommended to treat pneumonia in children [39].

### 4.4. Exclusion Criteria and Confounding Factors

Antibiotic use is dependent on age as older adults (aged > 65) use more antibiotics per capita than younger adults [28,29]. Furthermore, female sex is associated with high prescribing [29,31,40]. UDPs with missing values for gender and/or five-year age groups were removed. The five-year age groups were regrouped into 5 new groups: 15–24 years (adolescent), 25–44 (young adult), 45–64 (adult), 65–74 (senior), and 75+ (elder). Data on the total population size of each PC4 area by gender and five-year age category were collected from Statistics Netherlands for the year 2015 [41]. All PC4 areas with less than 100 inhabitants were excluded.

Commercial poultry and goat farms are seldom found in urban areas; however, other environmental risk factors, such as traffic-related air pollution, are present. PC4 areas that scored 1 (very strong, >2500 addresses per km^2^) or 2 (strong, 1500–2500 addresses per km^2^) on urbanization were therefore excluded. Urbanization scores per PC4 area were collected from Statistics Netherlands for 2014.

Multiple studies have shown the influence of socio-economic factors on the consumption of antibiotics [26,28,30]. The SES per PC4 area was collected from the Netherlands Institute for Social Research for the year 2014. SES was expressed as a status score based on population characteristics (educational level, income, and labor market participation) of each PC4 area, and was divided into quartiles: low (<−0.22), low-medium (≥−0.22 and <0.42), medium-high (≥−0.42 and <0.96), and high (≥0.96). PC4 areas with missing data on SES were excluded.

Smoking is associated with an increased risk of CAP as tobacco increases the susceptibility to bacterial lung infection [42,43]. The percentage of smokers per PC4 area was available from the RIVM health monitor [44] and was divided in quartiles: low (<17%), low-medium (≥17% and <19%), medium-high (≥19% and <21%), and high (≥21%).

### 4.5. Statistical Analysis

We used descriptive statistics to describe the characteristics of the population of UDPs with and without poultry or goat farms. For each antibiotic, we calculated the total number of prescriptions and the mean DDD per 1000 inhabitants. The amoxicillin, doxycycline, co-amoxiclav, and nitrofurantoin use, the presence of poultry and goat farms, and the SES distribution were visualized in maps using a Geographical Information System.

We first explored associations between antibiotic use and poultry and goat farm exposure variables in crude two-by-two associations using Chi-square analysis. In the main analyses, the number of DDDs per UDP, which is a count variable, was used as the outcome variable. Initial data exploration using Poisson regression showed that the ratio of deviance to degrees of freedom varied from 15.10 (amoxicillin) to 30.62 (doxycycline), indicating severe overdispersion of the outcome variable. To account for overdispersion we applied negative binomial regression analysis. Per type of antibiotic, two univariate negative binomial regression models were analyzed with the number of DDDs per UDP as outcome variable and the poultry farm or goat farm presence as determinant. In the multivariate negative binomial regression analyses, the effect estimates were adjusted for age, gender, SES, smoking, and poultry/goat farm presence. PC4 was added to all analyses as a random effect to correct for potential clustering of characteristics between PC4 areas. Population size was incorporated in the model with the use of the offset option, for which the population size variable had to undergo a log transformation. For each negative binomial regression model, estimates were exponentiated to calculate a risk ratio and 95% confidence intervals (CI). All statistical analyses were conducted in SAS 9.4 (SAS Institute Inc., Cary, NC, USA).

### 4.6. Sensitivity Analysis

Earlier research has shown peaks in CAP incidence during the winter months, due to certain etiological agents that show seasonal variation: Influenza virus, *Streptococcus pneumoniae*, and *Haemophilus influenzae* [37]. It is unclear whether there is also a seasonal effect in the relation between pneumonia and living in the vicinity of poultry and goat farms. We hypothesized that if an infectious agent transmitted from poultry or goat farms causes the elevated risk of pneumonia in humans and the agent itself has a specific seasonality, the effect estimates of the distance related associations will be stronger outside the flu season. To explore seasonal effects, we analyzed the negative binomial regression models stratified for the winter (November–March) and other months (April–October). Furthermore, we performed a sensitivity analysis on regional level by restricting our analysis to the most livestock dense areas in the Netherlands (provinces of Brabant, Limburg, Gelderland, Overijssel, and Utrecht).

## 5. Conclusions

This study suggests increased use of antibiotics among people living close to poultry farms but not goat farms. The study showed remarkable regional differences in antibiotic use which is only to a small extent explained by presence of poultry farms. Pneumonia is a very common condition, with approximately 270,000 episodes per year in the Netherlands. The vast majority of pneumonia patients are diagnosed based on clinical criteria in primary care and receive presumptive antibiotic treatment according to prevailing professional guidelines. A small increase in risk can therefore have a profound impact on disease burden and on the use of antibiotics. Disentangling the multiple determinants of antibiotic use, including presence of livestock, SES, smoking behavior, health seeking behavior, and prescribing practices in health care, will require complex individual-level studies.

## Figures and Tables

**Figure 1 antibiotics-10-01346-f001:**
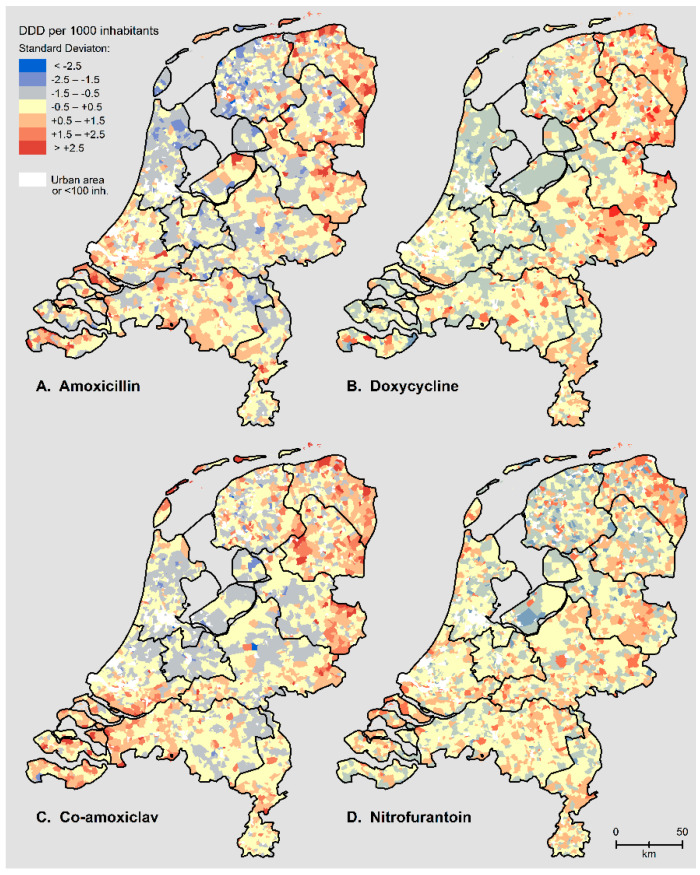
Antibiotic use (DDD per 1000 inhabitants) per 4-position Postal Code area (*n* = 2586) in the Netherlands in 2015 as the number of standard deviations from the mean. (**A**) Amoxicillin; (**B**) doxycycline; (**C**) co-amoxiclav; (**D**) nitrofurantoin. Urban areas and areas with fewer than 100 inhabitants are displayed in white.

**Table 1 antibiotics-10-01346-t001:** Population characteristics of Unique Data Points (UDPs) with and without poultry/goat farms.

Variable		Percentage of UDPs *			
		>0 Poultry Farms	0 Poultry Farms	>0 Goat Farms	0 Goat Farms
**Age**	15–24	13.0	13.1	13.0	13.1
	25–44	25.9	26.1	25.8	26.1
	45–64	26.1	26.4	26.0	26.3
	65–74	13.0	13.1	12.9	13.1
	75+	22.1	21.4	22.4	21.5
**Gender**	Male	49.6	49.7	49.7	49.7
	Female	50.4	50.3	50.3	50.3
**SES**	Low (<−0.22)	20.0	18.1	10.9	20.7
	Low-medium (≥−0.22 and <0.42)	32.1	23.0	33.7	25.1
	Medium-high (≥−0.42 and <0.96)	29.9	27.7	30.7	28.1
	High (≥0.96)	18.1	31.2	24.7	26.1
**Smoking**	Low (<17)	22.8	17.4	28.0	17.7
	Low-medium (≥17 and <19)	38.3	37.0	39.1	37.1
	Medium-high (≥19 and <21)	20.0	22.2	18.3	22.0
	High (≥21)	18.9	23.5	14.6	23.2
**Urbanization**	3 (1000–1500 addresses per km^2^)	10.8	22.5	10.7	19.4
	4 (500–1000 addresses per km^2^)	21.0	20.1	19.3	20.8
	5 (<500 addresses per km^2^)	68.2	57.4	67.0	59.9

* Percentage of PC4-age group-gender-month UDPs in the dataset (*n* = 993,024 UDPs). UDPs: Unique Data Points; SES: socio-economic status.

**Table 2 antibiotics-10-01346-t002:** Mean DDD per 1000 inhabitants in PC4 areas (*n* = 2586) by age group, gender, SES, smoking status, and presence of poultry or goat farms.

Variable		Mean DDD per 1000 Inhabitants			
		Amoxicillin	Doxycycline	Co-Amoxiclav	Nitrofurantoin
**Age**	15–24	33.9	55.4	27.1	28.5
	25–44	47.1	48.7	37.6	24.3
	45–64	55.3	64.9	48.5	34.6
	65–74	70.2	93.9	75.8	70.9
	75+	102.1	135.8	135.1	159.7
**Gender**	Male	59.0	78.8	72.6	19.5
	Female	65.8	78.4	57.8	105.8
**SES**	Low (<−0.22)	70.5	92.6	76.9	68.4
	Low-medium (≥−0.22 and <0.42)	65.2	83.9	68.7	66.6
	Medium-high (≥−0.42 and <0.96)	59.5	73.4	63.2	60.9
	High (≥0.96)	56.7	68.2	54.8	57.3
**Smoking**	Low (<17)	60.9	75.9	60.4	63.2
	Low-medium (≥17 and <19)	62.3	77.4	64.8	64.2
	Medium-high (≥19 and <21)	61.4	80.8	65.0	60.6
	High (≥21)	65.1	81.0	70.3	62.8
**Poultry farm presence**	Yes	65.2	84.8	68.2	65.2
	No	60.5	74.3	63.0	61.4
**Goat farm presence**	Yes	64.2	83.8	66.1	66.8
	No	62.0	77.4	64.9	62.0

DDD: Defined Daily Doses; PC4: 4-position postal code; SES: socio-economic status.

**Table 3 antibiotics-10-01346-t003:** Crude two-by-two associations per type of antibiotic with presence of poultry or goat farms.

Risk Factor	Amoxicillin RR (95% CI)	Doxycycline RR (95% CI)	Co-Amoxiclav RR (95% CI)	Nitrofurantoin RR (95% CI)
Presence of poultry farms	1.11 (1.11–1.12)	1.17 (1.17–1.18)	1.14 (1.14–1.15)	1.10 (1.09–1.11)
Presence of goat farms	1.14 (1.14–1.15)	1.15 (1.15–1.16)	1.15 (1.14–1.16)	1.15 (1.14–1.16)

RR: risk ratio; CI: confidence interval.

**Table 4 antibiotics-10-01346-t004:** Associations between presence of poultry or goat farms and antibiotic use per type of antibiotic from negative binomial regression.

**Poultry Farm Presence**	**Crude Ratio * (95% CI)**	**Adjusted Ratio ** (95% CI)**
Amoxicillin	1.08 *** (1.05–1.11)	1.05 (1.02–1.08)
Doxycycline	1.15 (1.11–1.20)	1.10 (1.06–1.14)
Co-amoxiclav	1.08 (1.05–1.12)	1.05 (1.02–1.08)
Nitrofurantoin	1.06 (1.02–1.11)	1.00 (0.96–1.04)
**Goat Farm Presence**	**Crude Ratio * (95% CI)**	**Adjusted Ratio ** (95% CI)**
Amoxicillin	1.05 (1.00–1.08)	1.02 (0.98–1.05)
Doxycycline	1.06 (1.01–1.12)	1.03 (0.98–1.08)
Co-amoxiclav	1.02 (0.98–1.06)	1.00 (0.96–1.03)
Nitrofurantoin	1.09 (1.03–1.15)	1.03 (0.98–1.09)

In all models, PC4 was included as random factor. CI: confidence interval; PC4: 4-position postal code. * Exponential of the estimate; ** Corrected for SES, age, gender, smoking, and presence of goat/poultry farms; *** Interpretation ratio: the amoxicillin use was 8% higher in the UDPs with poultry farms.

## Data Availability

The antibiotic prescription data that were analyzed during this study are available from the Drug Information System (GIP) of the National Health Care Institute (in Dutch: Zorginstituut Nederland) but restrictions apply to the availability of these data, which were used under license for the current study, and so are not publicly available.

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
