# Peer review of "Use of Antibiotics among Residents Living Close to Poultry or Goat Farms: A Nationwide Analysis in The Netherlands"

_antibiotics, 2021, doi:10.3390/antibiotics10111346_

Round 1
Reviewer 1 Report
There is no in-depth analysis of the specific factors in poultry farms that contribute to the increase in antibiotic use in the population around poultry farms, thus preventing appropriate control measures from being taken at the source.
Author Response
Thank you for the review of our manuscript. The livestock farming data that we used are location based, meaning that we only know the location, the type of livestock farm and the number of animals. We don’t have information about specific characteristics of these farms, such as free-range chickens, housing systems and type of air filtration systems. We have added this clarification in our revised manuscript in lines 258-260. Our epidemiological study adds evidence to the existing literature about possible health effects of local residents around livestock farms. Because of the ecological character, this study precludes inference about causality. Other (intervention) studies will have to show whether adjustments to livestock farming systems (e.g. emission reduction of particulate matter) will reduce the associations between living close to livestock farms and negative health effects.
Reviewer 2 Report
The paper is well presented and describes a study of associations between human antibiotic usage (of the type usually prescribed for respiratory disease) and proximity to livestock (goat and poultry farms). The study showed significant associations between such antibiotic use and proximity to livestock farms particularly poultry farms. There are several caveats to these findings which immediately come to mind of a reviewer but thankfully the authors appear to have considered most of these and factored them into their account. In particular of course no causal relationship can be inferred, but also the use of certain antibiotics as a surrogate for human respiratory disease, why 1km?; to name but three. A couple of minor observations:
lines 207-217. There are several statements in this paragraph e.g. use of nitrofurantoin and sexual distribution of urinary tract infections, which are unsupported by references. References would be helpful.
It took me a little while to figure out what PC4 and PC6 areas were. For those not familiar with the Netherlands' postal code system, a little more explanation would be helpful.
line 259: What was the justification for selection of minimal size of farms
Reviewer 3 Report
Dear Authors,
I would like to congratulate you on the article presented. I don`t have any specific recommendations or comments as I found the manuscript well organized and easy to read. My suggestion would be to put the method section before the results and discussion section.
Best Regards
Author Response
We thank you for your interest in our manuscript and we appreciate your compliments.